# ABC: Adversarial Behavioral Cloning for Offline Mode-Seeking Imitation Learning

**Eddy Hudson**[1]    **Ishan Durugkar**[1]    **Garrett Warnell**[1,2]    **Peter Stone**[1,3]

[1]UT Austin    [2]Army Research Laboratory    [3]Sony AI

`{eddy,ishand,pstone}@cs.utexas.edu`

`garrett.a.warnell.civ@army.mil`

## Abstract

Given a dataset of expert agent interactions with an environment of interest, a viable method to extract an effective agent policy is to estimate the maximum likelihood policy indicated by this data. This approach is commonly referred to as behavioral cloning (BC). In this work, we describe a key disadvantage of BC that arises due to the maximum likelihood objective function; namely that BC is mean-seeking with respect to the state-conditional expert action distribution when the learner's policy is represented with a Gaussian. To address this issue, we introduce a modified version of BC, Adversarial Behavioral Cloning (ABC), that exhibits mode-seeking behavior by incorporating elements of GAN (generative adversarial network) training. We evaluate ABC on toy domains and a domain based on Hopper from the DeepMind Control suite, and show that it outperforms standard BC by being mode-seeking in nature.

## 1   Introduction

While imitation learning (IL) is a powerful paradigm for skill learning, popular approaches such as Generative Adversarial Imitation Learning (GAIL) (Ho & Ermon, 2016) and Inverse Reinforcement Learning (Abbeel & Ng, 2004) require large amounts of data. Improvements to adversarial imitation learning (AIL) methods have made positive strides towards addressing this issue by leveraging advances in the sample complexity of RL algorithms and bringing them to bear in IL (Kostrikov et al., 2019; Hudson et al., 2021). Nevertheless, the problem of sample complexity in IL persists. Behavioral cloning (BC) is an exception as it is the rare IL algorithm that is also offline in nature. Aside from not requiring any additional interactions with the environment, BC is also a relatively straightforward algorithm. Given a dataset of state-action pairs that encapsulates interactions with an environment by a demonstrator, the problem of offline IL is reduced to a supervised learning problem, where a model is typically trained using maximum likelihood methods to predict the action taken by the demonstrator given an input state.

Unfortunately, the maximum likelihood objective function that is traditionally used in supervised learning leads to a shortcoming in BC when a Gaussian is used to model the state-conditional action distribution in the dataset, as is commonly the case for continuous control tasks. In such a scenario, the imitator will learn a Gaussian policy with a mean that coincides with the mean of this distribution. In many instances, predicting the mean will suffice. However, there are cases where predicting the mode would be preferable. Consider, for instance, a quadcopter that is faced with a tall tree that it can avoid by going around on either the left or right side of the tree, and assume the expert dataset contains demonstrations of both avoidance behaviors (Ke et al., 2020). A naive application of BC as described above would result in the quadcopter colliding with the tree as averaging the actions leading to the left and right side would result in an action that leads straight to the tree. Or consider the case of a continuous control problem where the dataset has been corrupted by the inclusion of

random actions from the uniform distribution. Adding the random actions shifts the mean while not affecting the mode, so predicting the mode would be advantageous.

Towards addressing these issues, we introduce a novel variant of BC called Adversarial Behavioral Cloning (ABC). ABC replaces the typical BC objective with a learned loss function as in generative adversarial networks (GANs) (Goodfellow et al., 2014). Using a bandit problem, we illustrate how ABC is mode-seeking while BC is mean seeking. We further evaluate ABC on a 2D navigation domain and in a problem setting based on Hopper from the DeepMind control suite.

## 2 Adversarial Behavior Cloning (ABC)

Our primary contribution in this work is an offline IL algorithm called Adversarial Behavioral Cloning (ABC) (Algorithm 1). ABC can be considered a hybrid of conditional GANs (Mirza & Osindero, 2014) and DDPG (Lillicrap et al., 2016). Based on the demonstration dataset, a discriminator is learned in the min-max style of GANs. This discriminator models the state-conditional action distribution in the dataset. We then train the policy by backpropagating through the discriminator (Line 6).

---

**Algorithm 1** Adversarial Behavioral Cloning (ABC)

---

1: Initialize parametric policy $\pi$, parametric discriminator $D(s, a)$
2: Obtain expert demonstration data $\tau_E = \{(s_t^*, a_t^*), (s_{t+1}^*, a_{t+1}^*), \ldots, (s_{t+n}^*, a_{t+n}^*)\}$
3: Initialize replay buffer $R = \{(s_i^*, a')\}$ with $N_R$ samples, where $i \in [0, n]$, $a' \in U(-1, 1)$ and $U$ denotes the uniform distribution.
4: Train $D$ to distinguish between $\tau_E$ and $R$
5: **for** $N$ iterations **do**
6:     Optimize $\pi$ by maximizing $\mathbb{E}_{i \in [0,n]}[D(s_i^*, \pi(s_i^*))]$
7:     Every $N_D$ iterations, add samples $\{(s_i^*, \pi(s_i^*))\}$ to $R$, and then train $D$ to distinguish between $\tau_E$ and $R$
8: **end for**

---

To illustrate how ABC works, and to show how it improves over BC, we devised a bandit problem where the agent is rewarded for picking an action value close to -1 or 1 (see the appendix for the precise definition of the reward function). The demonstration dataset is generated by sampling from a symmetric bimodal Gaussian (Figure 1a). In Line 3 of the algorithm, we seed the replay buffer ($R$) with samples from $U(-2, 2)$ instead. Due to the simplicity of the problem, we find that setting $N = 1$ suffices to achieve optimal performance (i.e., there is no need to add new samples to the replay buffer). As shown in Figure 1, ABC successfully reaches one of the modes in the distribution, while BC fails to learn one of the optimal action values by attempting to predict the mean in Figure 1a.

## 3 Experiments

We performed experiments in two domains: a 2D navigation domain and a locomotion domain. Both are based on example domains found in the DeepMind Control Suite (Tassa et al., 2018). The goal of our experiments in the 2D navigation domain is to show that ABC can successfully learn a meaningful policy when the expert's state-conditional action distribution contains multiple modes. Our experiments in the locomotion domain, on the other hand, serve to show that there can be instances where even when the distribution contains a single mode, it is advantageous to gravitate towards the mode.

### 3.1 2D navigation with a multimodal demonstration dataset

The 2d navigation domain is based on the point-mass domain from the Deepmind Control Suite. We designed it to be a testable representation of the quadcopter example discussed earlier. The agent in this task starts from the bottom left corner of the state space, and has to find its way to the top left corner or bottom right corner (see Figure 2a). At every timestep, the agent receives a reward as specified by the heatmap in Figure 2b. The episode terminates as soon as the agent comes into contact with the black square in the heatmap, thus forcing the agent to decide right at the beginning

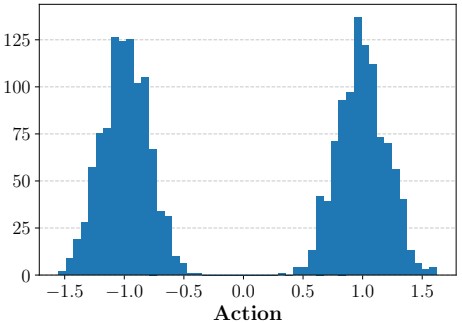

(a) Demonstration dataset. This was drawn from a bimodal Gaussian with modes at -1 and 1.

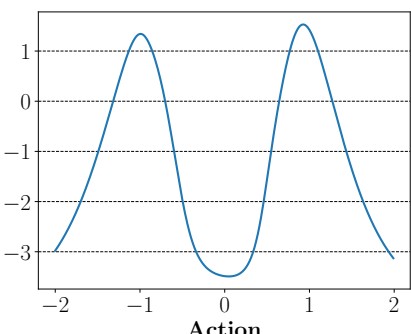

(b) Logit of the discriminator after executing line 5 of the algorithm.

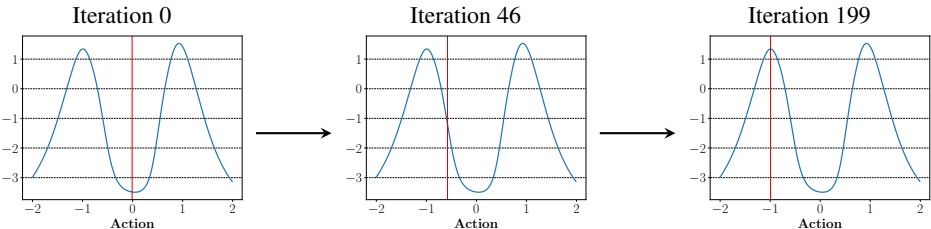

(c) Backpropagating through the discriminator to predict one of the modes (Line 7 of the algorithm). The policy is denoted by a red line.

Figure 1: The bandit problem illustrating how ABC works. While ABC is able to predict a mode of the distribution (1a), BC predicts the mean (-0.07), thus failing to achieve optimal reward.

which of the two targets it wishes to pursue. Unless the episode gets prematurely terminated in this manner, it stretches for a total of 1000 timesteps.

We produce a demonstration dataset for the top left target containing 1000 trajectories. In each of these 1000 trajectories, the agent begins from the bottom left corner and efficiently finds its way to the top left corner. We also produce a similar dataset for the bottom right corner. Combining these two datasets, we produce a third dataset with 2000 trajectories that contains data pertaining to both targets. As shown in Table 1, ABC successfully achieves a high reward in all three datasets. However, BC fails to obtain a meaningful reward in the combined dataset, when the state-conditional action distribution becomes multimodal due to there being two possible targets in the dataset.

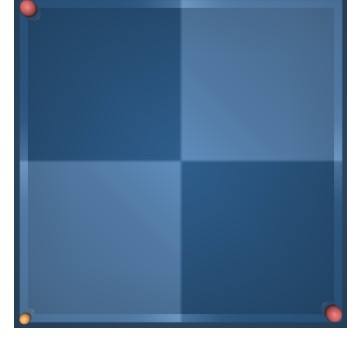

(a) The domain is based on the point mass agent from the DeepMind control suite.

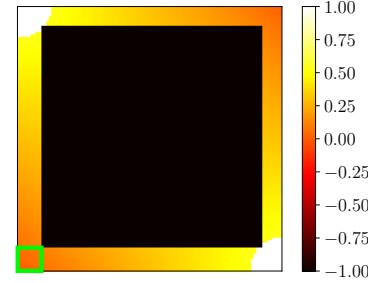

(b) Heatmap showing reward distribution in the state space. The episode ends if the agent enters the black square in the center. The agent starts each episode from a point in the green square.

Figure 2: The 2D navigation domain

|  | BC | ABC |
|---|---|---|
| **Top left (717)** | 758 | 778 |
| **Bottom right (708)** | 779 | 751 |
| **Combined** | 0.22 | 758 |

Table 1: Results in the 2D navigation domain. The left-most column contains the average reward of the trajectories in the dataset. BC and ABC results are the average of the last 10 points in the training curve. These results show that unlike ABC, BC fails when the state-conditional action distribution in the dataset is not unimodal.

## 3.2 Locomotion with corrupted data

Our experiments in this section were conducted in the Hopper domain from the DeepMind Control Suite (see Figure 3a), where the task is to move as quickly as possible (maximum reward 1000). We trained an expert agent using Soft Actor Critic (Haarnoja et al., 2018) and generated a dataset with 200 trajectories using the resulting expert policy, which obtained an average reward of 639. We also initialized a random policy and constructed a second dataset by using it to generate 100 trajectories which obtained an average reward of 0.39. Finally, we created a third, more difficult dataset by combining the two datasets described above. Let $s_d$ be the state where the random policy diverges from the expert policy. Since the random policy only contributes actions from the uniform distribution to $s_d$, the mode of the action at $s_d$ in this third dataset is unchanged compared to the mode in the first (expert) dataset. However, the mean is changed. Thus, a mean-seeking algorithm would fail to learn a good policy using this new dataset, while a mode-seeking algorithm would be unfazed by the change. As expected, ABC performs well in the task with the more difficult dataset, while BC fails to obtain meaningful reward (Figure 3b).

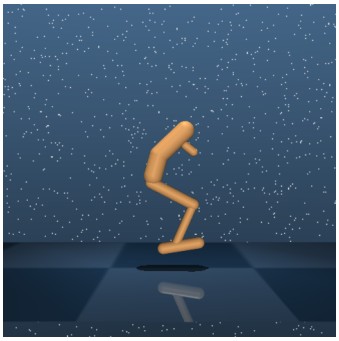

(a) The domain is based on the Hopper agent from the DeepMind control suite.

|  | BC | ABC |
|---|---|---|
| **Expert dataset** | 631 ($\pm$4) | 622 ($\pm$5) |
| **Expert+Random dataset** | 10 ($\pm$19) | 527 ($\pm$52) |

(b) Average reward. BC and ABC results are the average over 5 independent trials; standard deviations are shown in parentheses. BC fails on the inclusion of data from the random policy, while ABC does not.

Figure 3: The locomotion task with corrupted data

## 4 Conclusion

In this work, we identify a key disadvantage of BC that causes it to fall short when the demonstration dataset has state-conditional action distributions where it is more desirable to learn the mode instead of the mean. We address this shortcoming by proposing a novel variation of BC that we term Adversarial Behavioral Cloning (ABC). We evaluate ABC on multiple domains designed to exhibit the particular strengths of ABC, and show that BC indeed falls short.

An interesting future direction would be to evaluate ABC on more complicated variants of the 2D navigation problem (by perhaps replacing the point mass with the Ant agent from the DeepMind control suite). We are also interested in theoretically analyzing the objective optimized by the learned loss function, and understand how it relates to the reverse-KL, which is a mode-seeking loss function.

## 5    Acknowledgements

This work has taken place in the Learning Agents Research Group (LARG) at the Artificial Intelligence Laboratory, The University of Texas at Austin. LARG research is supported in part by the National Science Foundation (CPS-1739964, IIS-1724157, FAIN-2019844), the Office of Naval Research (N00014-18-2243), Army Research Office (W911NF-19-2-0333), DARPA, General Motors, Bosch, and Good Systems, a research grand challenge at the University of Texas at Austin. The views and conclusions contained in this document are those of the authors alone. Peter Stone serves as the Executive Director of Sony AI America and receives financial compensation for this work. The terms of this arrangement have been reviewed and approved by the University of Texas at Austin in accordance with its policy on objectivity in research.

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

## A    Appendix

### A.1    Reward function for the bandit

The reward function for the bandit is as follows:

$$M_r = 4 - \left[5(a-1)\right]^2$$
$$M_l = 4 - \left[5(a+1)\right]^2$$
$$R = \frac{1}{1 + e^{-M_r}} + \frac{1}{1 + e^{-M_l}}$$

Where $a$ is the action, and $R$ is the reward.

