# OpenReview forum: "ABC: Adversarial Behavioral Cloning for Offline Mode-Seeking Imitation Learning"
_NeurIPS.cc/2022/Workshop/Offline_RL — Offline RL Workshop NeurIPS 2022_

### Official Review · Reviewer_nmsU · 2022-10-10

**Rating:** 6
**Confidence:** 4

**Review:**

### Summary

The work studies the mean-seeking problem of Behavior Cloning (BC). A key disadvantage of maximizing likelihood using BC is that agents tend to fixate on the mean of dataset's state-conditioned action distribution. The work proposes a simple fix to this issue by designing a mode-seeking method called Adversarial Behavior Cloning (ABC). ABC learns a discriminator from demonstration dataset similar to conditional GANs. Gradients through the discriminator are backpropagated into the policy which is used to act and augment the dataset. Experiments in toy settings demonstrate ABC's mode-seeking behavior and improvement in task returns when compared to BC.

### Strengths

1. The paper builds clear intuition of why mode-seeking policies might be preferred in certain scenarios (eg- controlling a quadcopter).
2. Work is well written and clearly highlights experiment settings and their relevance.

### Weaknesses

1. A key aspect of the paper is the importance of mode-seeking policies. While mean-seeking policies have found a wide variety of applications, mode-seeking policies are seldomly used for multi-modal distributions. Mode-seeking agents such as ABC would collapse to a single mode when the number of modes is significantly large, hence ignoring the bulk of distribution's mass. It would be interesting if the authors could elaborate on the importance of mode-seeking policies in complex distributions.
2. Experiments demonstrate improved task returns for ABC where dataset possesses well-defined modalities in the distribution. Do these results generalize to other datasets where boundaries between different modes are skewed? How would other adversarial and implicit BC agents such as GAIL and Implicit Behavior Cloning (IBC) compare to ABC's mode-seeking approach?
3. This might be a minor comment, but I struggled to see the connection of ABC's algorithm design to this year's workshop theme. ABC restricts itself to the setting of learning dataset policies in a fully offline setting. It would be interesting if the authors could investigate its effects on downstream transfer to unseen tasks.

---

### Official Review · Reviewer_NCT2 · 2022-10-18

**Rating:** 3
**Confidence:** 4

**Review:**

The paper described a method for behavioral cloning based on adversarial learning.

The main issue with this method is that it has already been thoroughly studies in [1].

[1] Imitation Learning as f-Divergence Minimization
L. Ke, S. Choudhury, M. Barnes, W. Sun, G. Lee and S. Srinivasa
https://arxiv.org/abs/1905.12888